# Microstructure Evolution and Mechanical Properties of AZ80 Mg Alloy during Annular Channel Angular Extrusion Process and Heat Treatment

**DOI:** 10.3390/ma12244223

**Published:** 2019-12-16

**Authors:** Xi Zhao, Shuchang Li, Fafa Yan, Zhimin Zhang, Yaojin Wu

**Affiliations:** 1School of Mechanical and Electrical Engineering, North University of China, Taiyuan 030051, China; lishuchang_0001@163.com (S.L.); yanfafa_88@163.com (F.Y.);; 2School of Materials Science and Engineering, North University of China, Taiyuan 030051, China

**Keywords:** AZ80 Mg alloy, annular channel angular extrusion, heat treatment, microstructure, mechanical properties

## Abstract

Microstructure evolution and mechanical properties of AZ80 Mg alloy during annular channel angular extrusion (350 °C) and heat treatment with varying parameters were investigated, respectively. The results showed that dynamic recrystallization of Mg grains was developed and the dendritic eutectic β-Mg_17_Al_12_ phases formed during the solidification were broken into small β-phase particles after hot extrusion. Moreover, a weak texture with two dominant peaks formed owing to the significant grain refinement and the enhanced activation of pyramidal <c + a> slip at relative high temperature. The tension tests showed that both the yield strength and ultimate tensile strength of the extruded alloy were dramatically improved owing to the joint strengthening effect of fine grain and β-phase particles as compared with the homogenized sample. The solution treatment achieved the good plasticity of the alloy resulting from the dissolution of β-phases and the development of more equiaxed grains, while the direct-aging process led to poor alloy elongation as a result of residual eutectic β-phases. After solution and aging treatment, simultaneous bonding strength and plasticity of the alloy were achieved, as a consequence of dissolution of coarse eutectic β-phases and heterogeneous precipitation of a large quantity of newly formed β-phases with both the morphologies of continuous and discontinuous precipitates.

## 1. Introduction

As the lightest structural materials, Mg alloys attract constant attention in the aerospace, automobile, and electronics industries for their low density, high specific strength, and damping capacity [1,2]. Although Mg alloys exhibit many advantages, there are still some limitations in industrial applications owing to their low strength and poor formability. Therefore, to further expand the application range of Mg alloys, profuse efforts have been made to develop suitable thermo-mechanical processing methods for Mg alloys, aiming to improve the mechanical properties by refining their grains, as expressed by the Hall–Petch equation (σ_y_ = σ_0_ + kd^−1/2^, where σ_y_ is the yield strength, σ_0_ is the material constants, k is the Hall–Petch slopes, and d is the average grain size) [3]. Among them, severe plastic deformation (SPD) methods have aroused great interest in academic environments owing to their ability to significantly improve the mechanical properties of metal materials, such as equal channel angular process (ECAP) [4], accumulative roll bonding (ARB) [5], accumulative back extrusion (ABE) [6], multidirectional forging (MDF) [7], and cyclic expansion-extrusion (CEE) [8]. These techniques accumulate higher plastic strain values through multi-pass deformation, resulting in significant microstructure modification and greatly improved properties of Mg alloys. However, the biggest limitation is that the current SPD methods are only used for material properties improvement, but fail to prepare industrial-scale samples. In addition, the multi-pass forming process of most of these methods has the disadvantages of a long process cycle, cumbersome operation, and low production efficiency, which actually limit its further industrial application [9].

The key to promoting the application of Mg alloys as structural materials is to achieve coordinated control of material properties and forming processes. In recent years, annular channel angular extrusion (ACAE) was proposed as a combination of conventional back extrusion (BE) and ECAP methods [10]. Similar to the ECAP, this process introduced strong shear deformation and has the ability to refine the microstructure and weaken the basal texture concurrently. While improving the material properties, the process can form a cylindrical product at one time, greatly improving production efficiency. Hence, this seems to be a very promising SPD method for preparing high mechanical properties cylindrical parts that can be used in industry aspect. Finite element (FE) simulation by Shatermashhadi et al. [10] showed well that the obtained effective strain of the new method was more than twice that of the conventional BE method. Furthermore, Manafi et al. [11] introduced hydrostatic pressure in the ACAE to further increase the effective plastic strain and reduce the force load. Also, the feasibility of the process has been verified by the production of pure Al test part [12], and the extruded part achieved evident grain refinement in comparison with the initial material. However, these studies are still in the numerical simulation and small-scale part experiments stage, and have also not been successfully used in Mg alloy forming experiments.

In our previous work, the ACAE process was used to fabricate an AZ80 alloy shell part at 300 °C [13]. That study paid much attention to the effect of deformation behavior on material microstructure features, including the grain structure, phase morphology, and texture development, while the implication for heat treatment on the part has not been investigated. Besides, the lower deformation temperature did not result in fully dynamic recrystallized (DRXed) structure of the alloy, which was disadvantageous for further improvement in the mechanical properties. In this study, to clarify, the shell part of AZ80 Mg alloy was formed by the ACAE method at 350 °C with a slow punch of 0.5 mm/s. The effects of the new process on microstructure and different heat treatments on the precipitation mechanism as well as mechanical properties were analyzed specifically. Besides, investigations of the fracture behavior of the part in different conditions were realized using scanning electron microscopy (SEM).

## 2. Materials and Methods

In this investigation, a commercial as-cast AZ80 (Mg-8.0Al-0.5Zn-0.11Mn, wt %) alloy was employed. The alloy was provided in the form of a cylindrical ingot with a diameter of 100 mm and a height of 1000 mm. The as-cast bar was cut to 90 mm in diameter and 360 mm in length so as to be extruded. Prior to extrusion, the billet was preheated at the experimental temperature of 350 °C for 2 h to ensure the temperature consistency in the entire billet. Then, the extrusion was conducted at 4-THP-630 hydraulic press with a punch speed of 0.5 mm/s. Schematic of the ACAE process is illustrated in Figure 1. The movement of the punch caused the metal to flow through the bottom angular channel between the mandrel and die. Finally, a high quality shell part with 200 mm in outer diameter, 160 mm in inner diameter, and ~160 mm in height was successfully prepared, as shown in Figure 2a. Figure 2b shows the flow grid distribution obtained by our previous FE analysis [13] and sampling locations of the extruded part. The existence of two shear zones can effectively increase the effective strain through the accumulated shear strain. In order to evaluate the effect of shear strain on the microstructures, two typical regions (bottom and wall regions) were selected to investigate the microstructure evolution in detail. The heat treatment and tension test samples were selected from the wall region, and the specific heat treatment system selected in this paper is shown in Table 1.

The microstructure of the extruded sample was examined by optical microscopy (OM, A1m, Zeiss Inc., Oberkochen, Germany), scanning electron microscopy (SEM, SU5000, Hitachi Inc., Tokyo, Japan), and electron backscatter diffraction (EBSD, EDAX Inc., Mahwah, NJ, USA) installed in the SEM. Specimens for OM and SEM observation were mechanically polished and chemically etched in a solution of 0.4 g H_2_C_2_O_4_ + 1 mL CH_3_COOH + 60 mL distilled water for 5–10 s. Image-Pro Plus 5.0 software was used to statistic the grain size through OM maps. For EBSD examinations, the samples were ground and electro-polished in the electrolyte (90% C_2_H_5_OH + 10% HClO_4_) at 10 V for ~30 s and at −35 °C. EBSD was conducted with a scan step length of ~0.5–0.6 μm. The EBSD data were analyzed by orientation imaging microscopy (OIM) Analysis-v7.3 software (EDAX Inc., Mahwah, NJ, USA). In OIM software, the misorientation angle quantification was used to identify low angle grain boundaries (LAGBs, 2°–15°) and high angle grain boundaries (HAGBs, 15°–180°), as indicated in the EBSD maps by white and black lines, respectively. Besides, DRXed grains were identified by grain orientation spread (GOS) values smaller than 2°, while unDRXed grains had values greater than 2° [14]. The microhardness was determined using a Vickers indenter with a load of 200 g as well as a loading time of 15 s. To ensure the reliability of the results, 10 indentations were measured under each sample condition. The room temperature tensile tests were carried out on an Instron 5967 electronic tensile machine (Instron Inc., Canton, MA, USA) according to the ASTME8M-04 standard, with an constant strain rate of 1 × 10^−3^ s^−1^. Tension specimens, whose gauges were 15 mm in length and 4 mm in width, were extracted from the wall region, and then ground and polished to a final thickness of ~2 mm. To ensure that the test results were relatively accurate, at least three samples were tested in each group of experiments.

## 3. Results and Discussion

### 3.1. Initial Material State

Figure 3 shows the microstructure of the as-cast AZ80 Mg alloy specimen. It can be seen from this that the cast alloy consists mainly of α-Mg grains (~252 μm) and a small amount of Al_8_Mn_5_ compounds, with a macro-segregation of the intermetallic compounds (eutectic β-Mg_17_Al_12_ phases) at the grain boundaries [15]. Besides, there is a certain amount of lamellar-shaped and short-lath-shaped precipitates precipitated along the vicinity of eutectic phases. Previous studies have reported that the above two phases were discontinuous precipitate (DP) and continuous precipitate (CP) with the chemical component of Mg_17_Al_12_, respectively [16,17]. Generally, during solidification of the as-cast AZ80 alloy, the cooling speed is too fast to keep the uniform distribution of Al. Therefore, at the end of solidification, the remaining liquids are very rich in Al, and the microsegregation of Al content in the liquids further crystallizes into divorced eutectic phases. With the further cooling process, both DPs and CPs will also precipitate into the α-Mg matrix, especially in the vicinity of divorced eutectic phases owing to their high Al solute distributed [18].

### 3.2. Effect of Extrusion on Microstructure and Texture

Figure 4a–c show the microstructure from the bottom region of the extruded part. After passing through the shear zone 1, it can be seen that DRX was intensely developed and a large amount of fine DRXed grains formed along the initial grain boundaries. However, owing to the inadequate strain applied, the grain structure is characterized by a typical mixed form that composed of fine DRXed grains, relatively coarse DRXed grains, and residual elongated deformed grains. It is clear that in some areas with deformed grains and some grain boundaries, a certain amount of LAGBs have developed (Figure 4c and Figure 5a). In Figure 4b, it can be seen that the residual eutectic β-phases have also been significantly elongated and some fine β-phase particles were separated from them. It is obvious that fine DRXed grains in the varying size of ~8–12 μm have nucleated at these regions. This shows that, under plastic deformation, the strain incompatibility developed at the interface between the harder phase and matrix, and thus the high dislocation density accumulated around the phases provided sufficient energy storage for DRX nucleation, that is, the typical particle-stimulated nucleation (PSN) mechanism that has been reported frequently [19,20]. In addition, with the generation of the fine β-phase particles, the DRX grain boundaries’ migration in these regions was actually impeded to varying degrees, thereby inhibiting the growth of DRXed grains by the pinning effect [21]. When shifting to the wall region, it can be seen from Figure 4d–f that the deformed grains have been greatly consumed and the remaining eutectic β-phases were further elongated and significantly broken into fine β-phase particles, owing to further shear strain in shear zone 2. Clearly, the average grain size of extruded alloy was eventually refined to ~23.5 μm with relative uniform grain distribution. Compared with the bottom region, the grain boundary structure evolution presents that the LAGBs have been greatly reduced and converted to HAGBs (Figure 5b), indicating that adequate DRX was developed and a more complete DRX structure had formed. It is well known that LAGBs are proliferated by the consecutively climbing of dislocations, which can further absorb the moving dislocations and transform them into HAGBs, that is, the continuous DRX (CDRX) process [19]. Furthermore, in some serrated grain boundaries, obvious grain boundary bowing and fine grains nucleation can also be evidently observed (Figure 4c,f), which indicates that the traditional discontinuous DRX (DDRX) also appeared as a supplement. This is perfectly understandable because a relatively higher temperature (350 °C) promoted more pronounced atomic diffusion and dislocation migration toward the grain boundary regions. Overall, the current results yield a relatively complete and uniform DRX structure compared with our previous results [13] in processing the same material at 300 °C, despite the larger grain size obtained, and it is believed that this is very beneficial for subsequent heat treatment.

In Figure 6, the EBSD calculated grain orientation distribution (GOS) that distinguishes the DRXed grains (GOS < 2°) and unDRXed grains (GOS > 2°), as well as their relationship with the texture obtained from (0001)\discrete (0001) pole figures, are presented. It is found that significant grain-preferred orientations have occurred in these two regions and it should be noted that the texture development is concurrent with the DRX process. Clearly, the unDRXed grains (red and yellow grains) mainly enrich in the sides or adjacent to sides in the (0001) pole figures of the bottom region, whereas the DRXed grains are distributed around the unDRXed grains with a relatively random orientation. Hence, a strong basal texture occurred in the bottom region, which corresponds to the existence of high fraction of residual coarse grains with their basal planes aligned nearly parallel to the TD (red and yellow grains). After the second shear deformation, coarse original grains were continually consumed and rotated, and a large number of fine DRXed grains with relatively random orientation were generated. As a result, a weak DRX texture with multiple components and lower maximum basal pole density of ~7.8 was developed in the wall region. Extensive studies have clarified the relationship between DRX and texture evolution of Mg alloys [22,23,24]. Jia et al. [24] systematically studied the DRX and texture evolution of AZ31 Mg alloy in warm rolling, and well revealed the texture intensity weakening was inseparable with the improvement of DRX level resulted from rolling reduction. Similarly, maximum basal pole intensity decreased from ~29.6 to ~9.8 with the increment in DRX percentage in this result, which confirms the conclusion.

Besides, despite that significant weakening of the deformation texture was achieved in the wall region, it is worth noting that there are still two dominant texture components that have appeared, as marked in the figure. In our previous study [13], it was revealed that texture component 1 was actually a typical shear texture, which has also been widely reported in ECAP with various Mg alloys. However, the striking feature is that another distinct texture component already appeared with its basal poles inclined to ~0°–15° towards LD, which indicates that the higher temperature promoted the diversification of texture. Minarik et al. [25] studied the implication for different c/a ratio on the texture development of distinct Mg alloys processed by ECAP. They reported the similar texture component and clarified the enhanced activation of pyramidal <c + a> slip was responsible for the texture formation. In Mg alloys, it is well known that the critical resolved shear stress (CRSS) required for pyramidal <c + a> slip system at room temperature is ~100 times larger than that for the basal slip system. However, the possibility of pyramidal <c + a> slip activation is greatly increased at elevated temperatures owing to the fact that the activation of slip in the pyramidal plane is more energetically beneficial [26,27]. Hence, similar to the formation mechanism of shear texture, the pyramidal <c + a> slip promoted the rotation of the grain pyramidal planes aligned to the geometric shear plane [25], which can effectively split the typical shear texture into distinct parts, and thus greatly weaken the texture intensity, as shown in this extrusion process.

### 3.3. Effect of Heat Treatment on Microstructure

As is well known, AZ80 Mg alloy as a typical heat-treatable metal, and heat treatment plays a key role in improving its mechanical properties. Although some attempts have been made to break through the limitations of traditional heat treatment [28,29], the traditional method is still the main system for industrial production. In this work, two typical heat treatment systems were specifically implemented: direct aging treatment (T5) and solution treatment + aging treatment (T6). Referring to previous studies [28,30], a representative solution temperature of 415 °C and aging temperature of 175 °C were selected. Several groups of solid solution treatment (T4) experiments showed that the single α-Mg microstructure was almost completely obtained after 1.5 h of solid solution (not shown in this paper), and thus the solution time was terminated at 1.5 h. The corresponding age-hardening curves of the extruded alloy via these two heat treatment schemes are exhibited in Figure 7. This demonstrates that a significant age hardening response was obtained, where the hardness value increased to the peak-aged hardness of 91 HV and 103 HV for T5 and T6 samples, respectively. It is worth noting that the peak hardness and peak time of the T5 sample (24 h) are lower than those of the T6 sample (30 h). This indicates that the supersaturated solid solution formed after T4 treatment can provide sufficient solute atoms for subsequent aging, thereby significantly improving the hardening effect, but extending the peak aging time of the T6 sample.

Figure 8 shows the microstructure of the wall specimens after heat treatment by T4, T5, and T6, respectively. After T4 heat treatment, the initial extruded DRXed grains grew uniformly (~30 μm) and developed into the morphologies of equiaxed grain. It is clear that the banded eutectic β-phases as well as fine β-phase particles basically dissolved into the matrix. However, there is still a small amount of bulk Al_8_Mn_5_ intermetallic compounds remaining in the grain boundaries (Figure 8b). Hence, the Al_8_Mn_5_ intermetallic compound with high melting point was not involved during the T4 treatment [31]. Figure 8c shows the microstructure after T5 treatment. It can be seen that the banded eutectic β-phases still exist, while a large amount of the new formed β-phases have precipitated around the vicinity of eutectic β-phases and formed the precipitation band. This indicates that the eutectic β-phase regions contained a far higher Al content than the other matrix regions, and thus promoted the preferential precipitation of β-phases. The clearer phase morphology can be seen from the higher magnification SEM image, where the short-lath-shaped/fine granular CPs have precipitated at the front of lamellar DPs (Figure 8d). In general, lamellar-shaped DPs tend to preferentially precipitate in the grain boundaries with high Al content and extend into the grains. Precipitation of DPs leads to the drastic decrease in Al concentration of the matrix. When the Al content in the matrix is lowered to the critical value, the coarse lamellar-shaped DPs will be completely inhibited and the precipitated phases transform to be fine granular/short-lath-shaped CPs [17]. Braszczynska-Malik [16] also reported that the DPs and CPs occurred competitively at medium aging temperature (200 °C), and found the DPs were initialized at the grain boundaries, while the CPs distributed within the grains. After T6 treatment, a large amount of β-phases precipitated from the supersaturated solid solution. Figure 8e and f show that typical colonies of lamellar-shaped DPs always initiated at grain boundaries and grew perpendicular to the boundaries. Similar to the T5 sample, some short-lath-shaped CPs appeared at the front end of the lamellar-shaped DPs. However, it is noteworthy that complete CP areas have also appeared and occupied the recrystallized grain in the T6 sample where the DPs scarcely distributed, possibly owing to the lower Al content being insufficient to induce the precipitation of DPs.

### 3.4. Mechanical Properties and Fracture Morphology

Figure 9 presents the typical engineering stress–strain curves of extruded, T4, T5, and T6 specimens in the tensile directions of the longitudinal direction (LD) at room temperature. The tensile properties of all the samples and corresponding average grain size are summarized in Table 2. Three main trends can be seen from the tested results. Firstly, compared with the as-cast sample, the yield strength (YS) and ultimate tensile strength (UTS) of the extruded specimen were significantly improved by combining the strengthening effect of fine grains and β-phase particles. Secondly, the elongation (EL) to failure of T4 specimen dramatically increased to ~14%, but the YS dropped evidently owing to the grain coarsening and phases dissolution. Thirdly, compared with the extruded specimen, the YS and UTS of T5 and T6 specimens were increased as a result of precipitation strengthening of β-phases (DPs and CPs). Furthermore, T6 specimen showed the highest UTS of ~328 MPa with the appropriate elongation to failure of ~8.3%. Combined with the microstructure of each sample, the residual eutectic β-phases and the precipitated β-phases with the morphologies of DPs and CPs may be the main factors affecting the mechanical properties of the extruded alloy. In the T5 specimen, the new β-phases precipitated around the banded eutectic phases, which easily induced stress concentration and resulted in an unsatisfactory aging strengthening effect. This actually contributed to limited strength improvement, but greatly worsened the elongation of material. The eutectic β-phases were dissolved into the matrix and more equiaxed grains were developed by T4 treatment, which improved the elongation of the alloy. After T6 treatment, profuse lamellar DPs and a small amount of finer CPs precipitated along the grain boundaries and interiors, playing a vital role in the phase-hardening behavior by the Orowan mechanism [32,33], and thus significantly improving the mechanical properties of the extruded alloy.

To further clarify the performance difference of extruded alloy in different states, the SEM microstructure near the fracture surface of extruded, T4, T5, and T6 specimens is given in Figure 10. It is clear that the fracture positions of the as-extruded sample and the T5 sample were in the regions where the coarse eutectic β-phases are clustered (Figure 10a,c). In contrast, the T4 specimen was mainly broken along the grain boundaries (Figure 10b). Some previous studies have shown that brittle eutectic β-phases were the main cause of alloy fracture and low strength of AZ series Mg alloys [34,35]. Generally, at the beginning of the tensile test, dislocations tend to accumulate rapidly near large-sized eutectic β-phases, resulting in high dislocation pile-up and large stress concentration. Then, microcracks develop and expand in these regions, leading to premature fracture and deterioration of the elongation of material. Therefore, T4 heat treatment dissolved the residual eutectic β-phases, which enhanced the deformation coordination ability and enormously improved the elongation to failure of the extruded alloy. The location of microcracks in T6 specimen is different from that of other specimens, which were developed between the precipitated CPs/DPs and the Mg matrix (Figure 10d). Compared with the coarse eutectic β-phases, these dispersed lamellar DPs and short-lath-shaped\fine granular CPs can effectively prevent dislocation slip and enhance the strength of the material. However, it is well known that the interface of β-phase with Mg matrix is not coherent, and the β-phase cannot be sheared by dislocations during the deformation process [36]. As a result, the interface between Mg matrix and β-phase is still the preferential position for crack nucleation. During tensile deformation, when the localized stress exceeds the critical value, microcracks tend to nucleate and propagate around these regions, resulting in the final material fracture. The above evidence indicates that extrusion and solid solution are important means of breaking and dissolving the brittle eutectic β-phases, which is very helpful for the mechanical properties improvement of the alloy after further aging treatment.

## 4. Conclusions

In this study, the AZ80 Mg alloy shell part was formed by the ACAE process at 350 °C. The effects of extrusion and heat treatment on the microstructure and mechanical properties of the part were systematically studied. The obtained results are summarized as follows:(1)After ACAE pressing, DRX of Mg grains was developed and the network eutectic β-phases formed during the solidification were broken into small β-phase particles. Moreover, weak texture with two dominant peaks formed owing to the significantly grain refinement and the enhanced activation of pyramidal <c + a> slip at relative high temperature.(2)T4 treatment caused the dissolution of residual β-phases and the development of more equiaxed grains. After the T5 process, β-phases that both DPs and CPs were preferentially initiated at banded eutectic phases owing to high distributed Al content. The T6 heat treated samples obtained remarkable age hardening response with much more lamellar-shaped DPs and granular/short-lath-shaped CPs precipitated.(3)The residual eutectic β-phases were the main cause of the fracture of the alloy. T4 treatment greatly increased the elongation of the alloy owing to dissolution of the residual eutectic β-phases. After T5 treatment, the tensile properties of the alloy increased as the expense of elongation. The highest material YS of ~235 MPa and UTS of ~328 MPa were achieved after T6 treatment with appropriate elongation to failure of ~8.3%.

## Figures and Tables

**Figure 1 materials-12-04223-f001:**
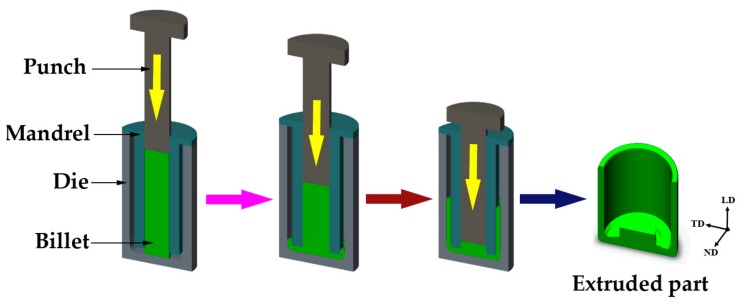
Schematic view of the annular channel angular extrusion applied in this research (TD, transverse direction; ND, normal direction; LD, longitudinal direction).

**Figure 2 materials-12-04223-f002:**
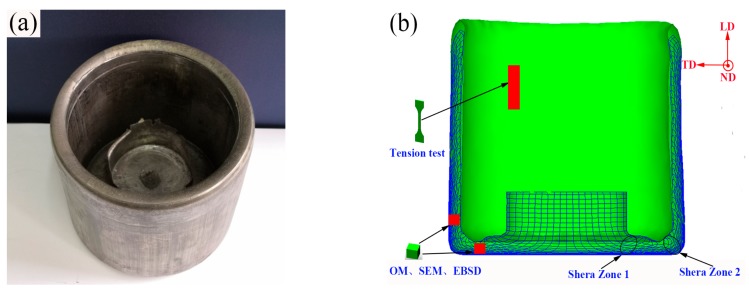
(**a**) The product of the extruded part, and (**b**) the flow grid distribution and corresponding sampling locations were used in this research. OM, optical microscopy; SEM, scanning electron microscopy; EBSD, electron backscatter diffraction.

**Figure 3 materials-12-04223-f003:**
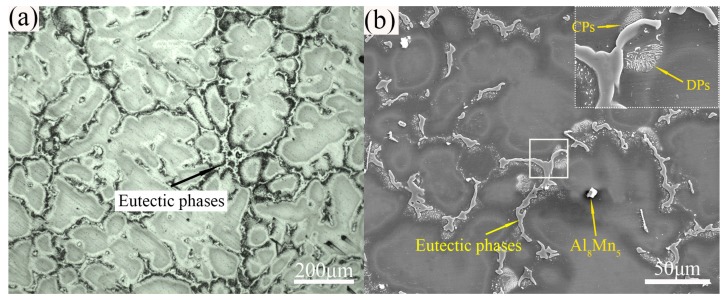
Microstructure of the as-cast AZ80 Mg alloy: (**a**) optical microscopy (OM) and (**b**) scanning electron microscopy (SEM).

**Figure 4 materials-12-04223-f004:**
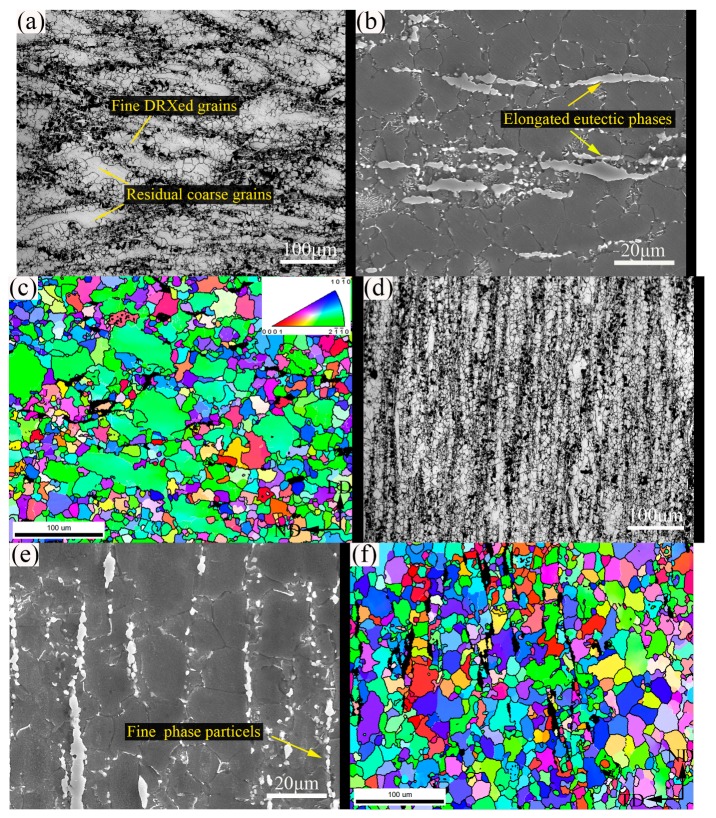
(**a**,**d**) OM, (**b**,**e**) SEM, and (**c**,**f**) electron backscattering diffraction (EBSD) apparatus figures of (**a**,**c**) the bottom region and (**d**,**f**) the wall region. DRXed, dynamic recrystallized.

**Figure 5 materials-12-04223-f005:**
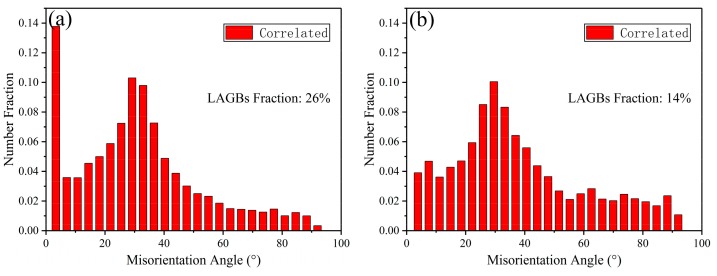
Misorientation distributions of the extruded AZ80 alloy shell part at (**a**) the bottom region and (**b**) the wall region. LAGBs, low angle grain boundaries.

**Figure 6 materials-12-04223-f006:**
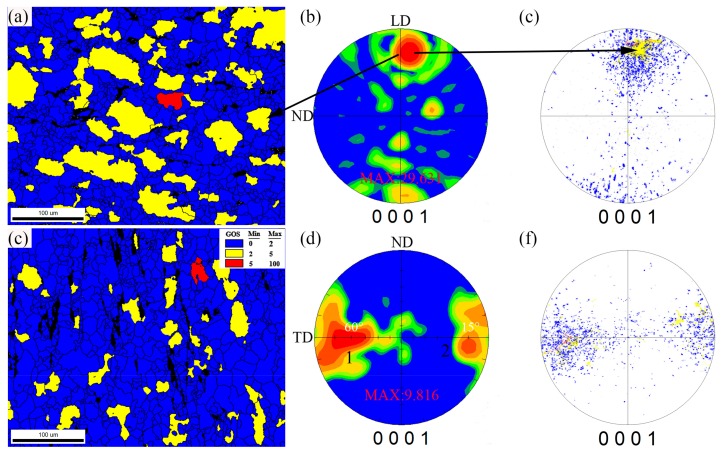
(**a**,**c**) EBSD maps distinguishing the full dynamic recrystallized (DRXed) zones (blue areas), unDRXed zones (yellow and red areas), (**b**,**e**) corresponding (0001) pole figures, and (**c**,**f**) discrete (0001) pole figures from EBSD data.

**Figure 7 materials-12-04223-f007:**
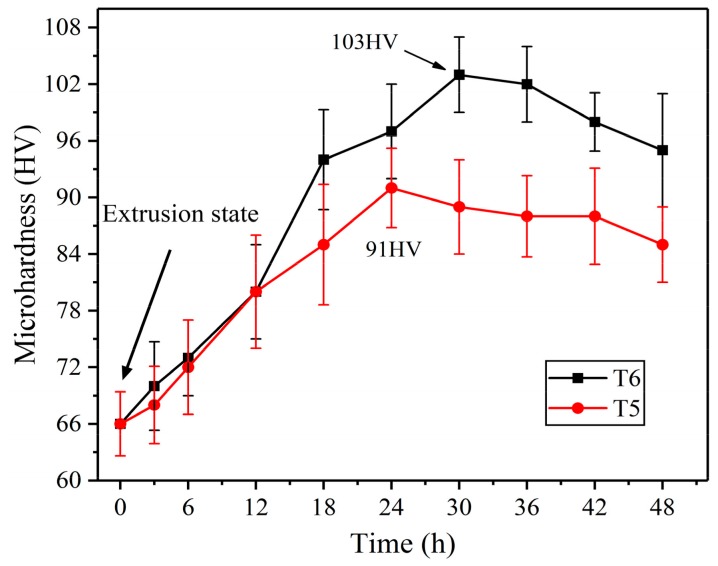
The age-hardening curves of the extruded AZ80 alloy samples via direct aging treatment for 24 h (T5), and solution treated at 415 °C for 1.5 h (T4) + aging at 175 °C for 30 h (T6).

**Figure 8 materials-12-04223-f008:**
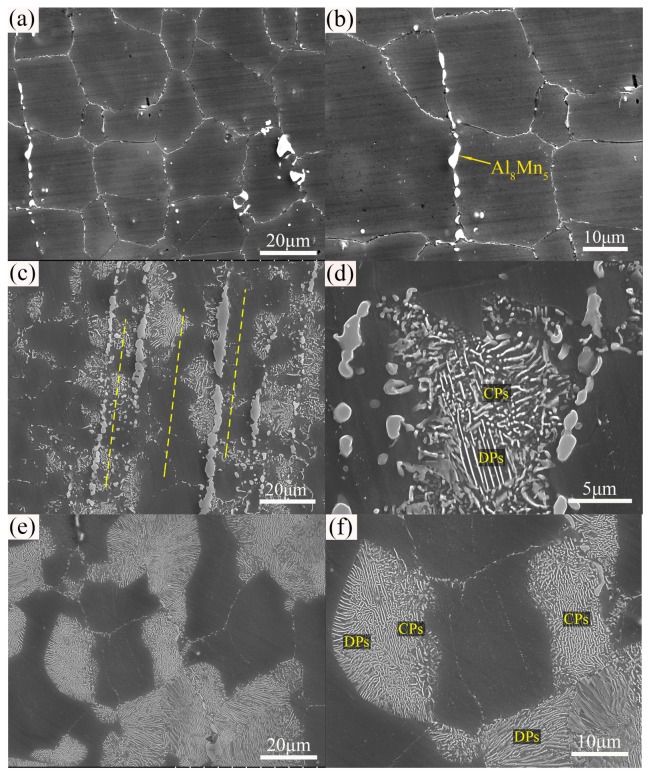
SEM microstructures of extruded AZ80 Mg alloy heat treated by: (**a**,**b**) T4, (**c**,**d**) T5, and (**e**,**f**) T6. CP, continuous precipitate; DP, discontinuous precipitate.

**Figure 9 materials-12-04223-f009:**
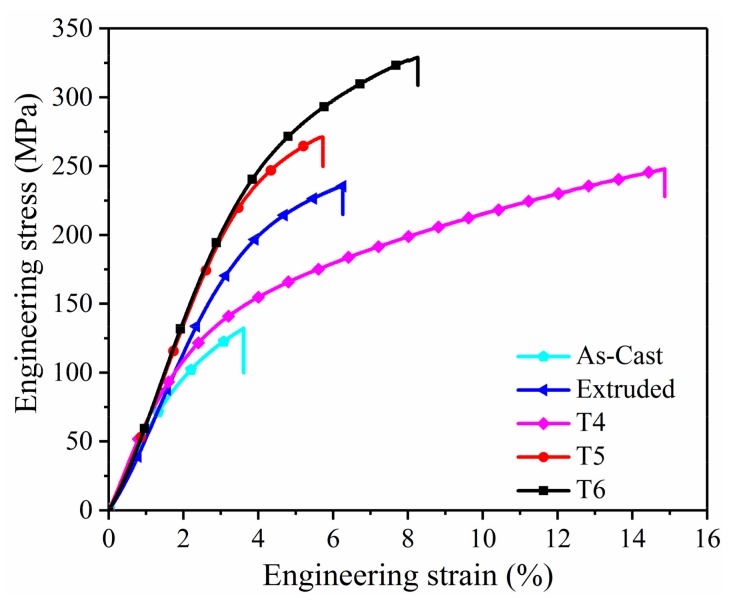
Engineering stress vs. strain curves of the extruded AZ80 alloy with different states.

**Figure 10 materials-12-04223-f010:**
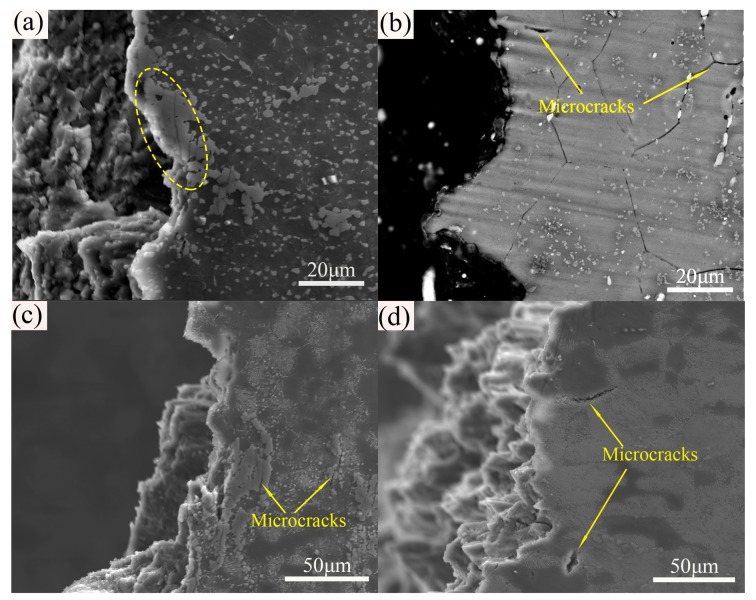
SEM microstructures of near fracture surface for extruded AZ80 alloy at different states: (**a**) extruded, (**b**) T4, (**c**) T5, and (**d**) T6.

**Table 1 materials-12-04223-t001:** Heat treatment schedule for the extruded AZ80 alloy.

Designation	Heat Treatment Schedule
T4	Solution treated at 415 °C for 1.5 h
T5	Aging at 175 °C for 24 h
T6	Solution treated at 415 °C for 1.5 h, aging at 175 °C for 30 h

**Table 2 materials-12-04223-t002:** The results of tensile tests at room temperature, and measured average grain size at different states of the extruded AZ80 Mg alloy. YS, yield strength; UTS, ultimate tensile strength; EL, elongation.

States	d_ave__rage_ (μm)	YS (MPa)	UTS (MPa)	EL (%)
As-cast	252 (±7)	71 (±5)	132 (±4)	3.6 (±0.5)
Extruded	23.5 (±2.6)	182 (±4)	238 (±5)	6.3 (±0.8)
T4	30.0 (±3.2)	118 (±2)	248 (±3)	14 (±0.5)
T5	25.8 (±3.2)	212 (±5)	270 (±6)	5.7 (±1.0)
T6	31.1 (±3.5)	235 (±4)	328 (±5)	8.3 (±0.7)

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
