# Peer review of "Microstructure Evolution and Mechanical Properties of AZ80 Mg Alloy during Annular Channel Angular Extrusion Process and Heat Treatment"

_materials, 2019, doi:10.3390/ma12244223_

Round 1

Reviewer 1 Report

Nice paper and useful findings, from time to time a work like this give good results.

Several points to make it even better:

Figures about textures are very good, please give more details….do you have other values about roughness, microindentations, etc.?

You go very rapidly to Methods. Introduction and state of the art can be improve.

Testing; extend explanations, because there is a great controversy today. . In the recent past there were researchers taking into account the effect of manufacturing method to obtain testpieces and coupons, showing effects important for the testing campaigns. The International society of experimental mechanics (SEM) has proposed in several work in the journal experimental techniques, in which for instance is the work, https://doi.org/10.1007/s40799-016-0134-5 was focused on tensile tests, but other dealt with the same idea, such as: https://doi.org/10.1007/s40799-016-0058-0, and https://doi.org/10.1016/j.matdes.2011.03.049. All make some ideas complementing this standards, such as ASTM E 8M-04, Standard test methods for tension testing of metallic materials

Author Response

Dear reviewer:

Thank you for your letter and for the comments concerning our manuscript entitled “Microstructure Evolution and Mechanical Properties of AZ80 Mg Alloy during Annular Channel Angular Extrusion Process and Heat Treatment” (materials-647812). The comments are all valuable and very helpful for revising and improving our paper, as well as the important guiding significance to our research. We have studied comments carefully and have made correction which we hope meet with approval. Revised portion were marked in red, as follows:

Point 1: Figures about textures are very good, please give more details….do you have other values about roughness, microindentations, etc.?

Response 1: Our main goal in this paper is to study the effect of this new extrusion process on the microstructure evolution as well as subsequent heat treatment on the tensile property and fracture behavior of the extruded material. And this is only a preliminary and basic study of this extrusion technology. Therefore, we did not consider other tests such as microindentation, corrosion resistance, fatigue property and so on. In view of your reminder, other material performance testing experiments may be considered in our future work.

Point 2: Testing; extend explanations, because there is a great controversy today. In the recent past there were researchers taking into account the effect of manufacturing method to obtain test pieces and coupons, showing effects important for the testing campaigns. The International society of experimental mechanics (SEM) has proposed in several work in the journal experimental techniques, in which for instance is the work, https://doi.org/10.1007/s40799-016-0134-5 was focused on tensile tests, but other dealt with the same idea, such as: https://doi.org/10.1007/s40799-016-0058-0, and https://doi.org/10.1016/j.matdes.2011.03.049. All make some ideas complementing this standards, such as ASTM E 8M-04, standard test methods for tension testing of metallic materials.

Response 2: Thank you very much for your comments on tensile testing, and we have read the related articles you sent. Before this, we have also done some experiments on the design of tension sheet, including the width, thickness and transition fillet of tension sheet, aiming to improve the stability and reliability of test results. In this study, the room temperature tensile tests were carried out on an Instron 5967 electronic tensile machine, and it is believed that there is not too much machine-induced test error. Although the current tensile test specimens are not standard specimens, all tensile tests are carried out in accordance with ASTME8M-04 standard. A slight change regarding the tensile performance test has been added to the method section, please check (Lines 122-126).

Once again, thank you very much for your comments and suggestions.

Best regards

Sincerely yours

Xi Zhao

Reviewer 2 Report

The aim of this work, treating on evolution of microstructure and mechanical properties of one of the most important magnesium alloy, AZ80, during annular channel under heat treatment, has been achieved, and the paper can be considered for publication.

   As to the merit, Figure 1, in the scheme,  should be corrected by adding the symmetry axis, to show that it is not a flat but circular part considered. Moreover, concerning the edition, English should be improved before this work is approved as sufficiently right.

   The authors seem to not differentiate between Adverb and Adjective in their writing. Some sentences (e.g. as indicated by color markings in the pdf Enclosure) are not complete, so they should be corrected.

   The Discussion part is very short, and there is no Conclusion in this paper.  

Author Response

Dear reviewer:

Thank you for your letter and for the comments concerning our manuscript entitled “Microstructure Evolution and Mechanical Properties of AZ80 Mg Alloy during Annular Channel Angular Extrusion Process and Heat Treatment” (materials-647812). The comments are all valuable and very helpful for revising and improving our paper, as well as the important guiding significance to our research. We have studied comments carefully and have made correction which we hope meet with approval. Revised portion were marked in red, as follows:

Point 1: As to the merit, Figure 1, in the scheme, should be corrected by adding the symmetry axis, to show that it is not a flat but circular part considered

Response 1: The redrawn picture has replaced the original photo, please check.

Point 2: Concerning the edition, English should be improved before this work is approved as sufficiently right. The authors seem to not differentiate between Adverb and Adjective in their writing. Some sentences (e.g. as indicated by color markings in the pdf Enclosure) are not complete, so they should be corrected.

Response 2: We are very sorry for our incorrect writing in this paper. As you have mentioned, some sentences have indicated by color markings in the PDF enclosure, but this enclosure has not been sent to us. Hence, is it convenient to provide these comments again? Besides, we have modified some sentences due to the duplicate matching requirements and potential errors, please check.

Point 3:  The Discussion part is very short, and there is no Conclusion in this paper.  

Response 3: We are very sorry to write the conclusions wrongly into the discussion. The discussion and results of this article already were combined in a section. We have corrected the error, please check it.

Once again, thank you very much for your comments and suggestions.

Best regards

Sincerely yours

Xi Zhao
